# Relevance of Neurotrophin Receptors CD271 and TrkC for Prognosis, Migration, and Proliferation in Head and Neck Squamous Cell Carcinoma

**DOI:** 10.3390/cells8101167

**Published:** 2019-09-28

**Authors:** Yannick Foerster, Timo Stöver, Jens Wagenblast, Marc Diensthuber, Sven Balster, Jennis Gabrielpillai, Hannah Petzold, Christin Geissler

**Affiliations:** Department of Otolaryngology, University Hospital Frankfurt, 60590 Frankfurt am Main, Germany; yannick@foerster-wiesbaden.de (Y.F.); Timo.stoever@kgu.de (T.S.); Jens.wagenblast@kgu.de (J.W.); Marc.diensthuber@kgu.de (M.D.); sven.balster@kgu.de (S.B.); jennis.gabrielpillai@kgu.de (J.G.); Hannah.petzold@kgu.de (H.P.)

**Keywords:** HNSCC, CD271, p75, Trk, TrkC, HPV, LM11A 31, PF-06273340, neurotrophins

## Abstract

Head and neck squamous cell carcinoma (HNSCC) is the sixth most common cancer worldwide and often has a poor prognosis. The present study investigated the role of the low affinity nerve growth factor receptor CD271 as a putative therapy target in HNSCC. Neurotrophins that bind to CD271 also have a high affinity for the tropomyosin receptor kinase family (Trk), consisting of TrkA, TrkB, and TrkC, which must also be considered in addition to CD271. A retrospective study and functional in vitro cell line tests (migration assay and cell sorting) were conducted in order to evaluate the relevance of CD271 expression alone and with regard to Trk expression. CD271 and Trks were heterogeneously expressed in human HNSCC. The vast majority of tumors exhibited CD271 and TrkA, whereas only half of the tumors expressed TrkB and TrkC. High expression of CD271-positive cells predicted a bad clinical outcome of patients with HNSCC and was associated with distant metastases. However, the human carcinomas that also expressed TrkC had a reduced correlation with distant metastases and better survival rates. In vitro, CD271 expression marked a subpopulation with higher proliferation rates, but proliferation was lower in tumor cells that co-expressed CD271 and TrkC. The CD271 inhibitor LM11A 31 suppressed cell motility in vitro. However, neither TrkA nor TrkB expression were linked to prognosis or cell proliferation. We conclude that CD271 is a promising candidate that provides prognostic information for HNSCC and could be a putative target for HNSCC treatment.

## 1. Introduction

Head and neck squamous cell carcinoma (HNSCC) is the most common cancer entity of the upper aerodigestive tract and the sixth most common cancer worldwide [1]. HNSCC is associated with lower survival rates and is responsible for the second highest cancer-related mortality rate [2]. Cancer relapse and the appearance of distant metastases worsen the prognosis. Approximately 10% of N0–N1 carcinomas and 30% of N2–N3 tumors form distant metastases [3]. Due to such poor prognoses of patients with HNSCC, the search for new treatment strategies is imperative. CD271 (cluster of differentiation 271) is a transmembrane protein (gene encoded on 17q21.33) belonging to the tumor necrosis factor receptor superfamily, which plays both positive and negative roles in the development of several types of cancer [4,5,6]. Neurotrophins, including nerve growth factor (NGF), brain-derived-neurotrophic factor (BDNF), and neurotrophin-3/4 (NT-3/4), which share a sequence homology of approximately 50%, bind to CD271 with low nanomolar affinities [7,8,9]. The extracellular domain of CD271 contains a cysteine-rich region with a binding site for neurotrophins. Ligand binding recruits distinct cytoplasmatic adapter proteins for downstream signaling [10]. CD271 signaling pathways are very variable depending on the different tissues that they operate in and the ligands that bind to the receptor, but recent studies have shown that CD271 could determine the migratory and proliferative properties in melanoma and HNSCC through the activation of Ras homolog gene family member A (RhoA), Jun kinase (JNK), and nuclear factor kB (NF-kB) pathways [11,12,13,14,15,16,17]. CD271 was shown to mark a highly proliferative subpopulation in HNSCC [5,12,18], with expression of the receptor gradually increasing alongside the appearance of distant metastases in melanoma [11]. On the other hand, neurotrophins bind to a second class of receptors, the Trk receptors (tropomyosin receptor kinase), with higher picomolar affinity. The Trk gene family includes TrkA (1q23.1), TrkB (9q21.33), and TrkC (15q25.3), and each protein is heterogeneously expressed in the pharyngeal mucosa and in different types of cancer, including HNSCC [19,20]. Trk receptors are tyrosine kinases, whereby the extracellularly located second immunoglobulin-like domain determines ligand-binding specificity. Unlike the tyrosine kinase domains, which share a protein homology of approximately 80%, the extracellular domains are less analogous (30% homology), resulting in different ligand-binding affinities. TrkA binds to NGF, while BDNF binds to TrkB and NT3 binds to TrkC. Ligand binding triggers different intracellular pathways and leads to the activation of mitogen-activated kinase (MAPK), phospholipase Cγ (PLCγ) and phosphoinositide 3-kinase (PI3K) cascades [10]. The Trk receptors and CD271 show no sequence homology; they bind to different neurotrophin regions, which is why they can simultaneously bind nerve growth factors, form heterodimers, and work together [10]. Consequently, the function and intracellular signaling may vary depending on CD271 and Trk co-expression. For instance, CD271 activation instigates cell apoptosis in TrkA-negative cells, but promotes cell survival if TrkA is co-expressed [7,9,21,22]. Despite these findings, the exact functions of CD271 and Trk receptors remain unknown in HNSCC. As CD271 expression seems to be correlated with the appearance of distant metastases in melanoma and enhanced proliferation and migratory properties in HNSCC, we speculated that CD271 could be related to clinical outcome and the presence of distant metastases in HNSCC, respectively. Hence, we evaluated expression patterns of CD271 with regard to clinical data, including the presence of metastases, tumor stage, tumor localization, human papillomavirus (HPV) status, and survival rate in human primary HNSCC. Expression patterns of Trk receptors alone and in combination with CD271 were also considered. Further, the relevance of CD271 for cell proliferation and migration was investigated in vitro through the use of flow cytometry, and Trk expression was also evaluated. Moreover, we assumed that proliferation and migration would be affected by stimulation or inhibition of neurotrophin receptors in vitro.

## 2. Materials and Methods

### 2.1. Patients

A retrospective study conducted at the University Hospital Frankfurt am Main (Germany) was part of this investigation. The study was approved by the Ethics Committee of the Department of Medicine at the Johann Wolfgang Goethe University, Frankfurt am Main, Germany (464/17). For this explorative data analysis, primary HNSCCs of 184 patients were included. Tissue samples were collected between 1997 and 2008 during surgical resection and frozen directly at −80 °C. The patient population is described in Table 1. All disposable information of the patients was analyzed according to the expression patterns of CD271, TrkA, TrkB, and TrkC. No implications of missing values were done. Sex and year of diagnosis were not considered. The five-year overall survival (OS) indicated the survival rate within five years after diagnosis. Five-year disease-free survival (DFS) described the length of time without any symptoms of the disease. Tumor size (T stage), lymph node metastasis (N stage), distant metastasis (M stage), grading (G stage), and UICC (Union Internationale Contre le Cancer) staging were done in accordance with the 7th edition of the AJCC (American Joint Committee on Cancer) cancer staging manual. Patients were treated with surgery, single radio(chemo)therapy (R(C)T, (radiochemotherapy or radiotherapy), or a combination of both (surgery + R(C)T). Therapy decisions were made depending on the UICC stage and patients’ comorbidities. Carcinomas exhibiting keratin pearls were classified as keratinized. The determination of HPV-mediated tumors is indicated below.

### 2.2. Sample Preparation and Immunohistochemistry

Frozen tissue samples were prepared and cut into 7-μm-thin sections using CryoStar NX50 microtome (ThermoFisher, Waltham, MA, USA). Before the staining procedure, sections were fixed with methylene alcohol at a temperature of −20 °C for 5 minutes and washed with washing buffer (WL583C2500; DCS, Hamburg, Germany). The blocking of endogenous peroxidase was performed by incubating the sections with dual endogenous enzyme block (S2003; Dako, Carpinteria, CA, USA) for 10 minutes. Primary monoclonal antibodies were diluted in the antibody diluent (S3022; Dako): LNGFR (1/300, ab3125; Abcam, Cambridge, UK), TrkA (1/400, 2510; Cell Signaling, Cambridge, UK), TrkB (1/200, 4607; Cell Signaling) and TrkC (1/1000, 3376; Cell Signaling). The sections were incubated for 1 hour at room temperature. In the next step of the procedure, a detection line staining kit (PD000RP; DCS, Hamburg, Germany) was used according to the manufacturer’s instructions. The staining was developed for 5 minutes using DAB (3-3’-Diaminobenzidine) reagent (DC137C100; DCS, Hamburg, Germany) After washing and counterstaining with hematoxylin (A4840; AppliChem, Darmstadt, Germany), the sections were dehydrated with a series of ethanol dilutions (70%, 95%, and 100%) and xylene. Pictures were captured using a Zeiss Axioplan 2 (AxioCam ICC1 camera; Zeiss, Oberkochen, Germany). The staining was classified by the number of positive tumor cells. Tumor samples with more than 10% positive cells were considered to be positive for a particular receptor (CD271+, TrkA+, TrkB+, or TrkC+). Tumors with less than 10% positive cells were considered to be receptor negative samples (CD271−, TrkA−, TrkB−, or TrkC−).

### 2.3. Immunofluorescence

After fixation with methylene alcohol, sections were incubated with PBT (1 × Dulbecco’s phosphate-buffered saline (DPBS), 5% goat serum, 1% bovine serum albumin (BSA), and 0.1% TritonX) for 30 minutes, followed by incubation with primary antibodies diluted in PBT1 (TrkA 1/80, TrkB 1/200, and TrkC 1/1000) overnight at 4 °C. Secondary antibodies anti-rabbitTRITC (Tetramethylrhodamine) and anti-mouseFITC (Fluorescin isothiocyanate) diluted in PBT (1/200) were applied and incubated for another hour at room temperature. Pictures were taken with a Zeiss Axio Imager M2 after counterstaining with DAPI (4’,6-diamidino-2-phenylindole) diluted in DPBS (1/1000; 14190-094; Gibco, Waltham, MA, USA).

### 2.4. Determination of HPV-Mediated Carcinomas

HPV-mediated carcinomas were determined by general HPV DNA detection, HPV DNA sequencing, and p16 detection. The p16 expression intensities and patterns were evaluated according to the eighth edition of the cancer staging manual for oropharyngeal HNSCC. This classification was used for the oropharynx, the hypopharynx, and the larynx. Cases that were positive for HPV DNA and fulfilled the p16 expression criteria (nuclear and cytoplasmic expression in ≥75% of tumor cells with medium/high intensity of p16 expression) were classified as HPV-mediated cases. The most prominent HPV type was HPV-16 (18 cases). One carcinoma determined by HPV-18 was detected (1 case).

#### 2.4.1. DNA Isolation

The tumor material was taken in a standard surgery procedure and kept frozen at −80 °C. The samples were crushed by TissueLyser LT (Qiagen, Hilden, Germany). DNA purification was performed with the QIAamp DNA Mini Kit (Qiagen, Hilden, Germany) according to the manufacturer’s instructions. DNA was eluted in elution buffer.

The DNA concentration and purity were measured with ultraviolet cuvettes in a Helios alpha spectrometer (Spectronic Unicam, Leeds, UK) at wavelengths of 280 nm and 260 nm.

#### 2.4.2. Polymerase Chain Reaction

The detection of a β-globin gene (5′-ACA CAA CTG TGT TCA CTA GC-3′, 5′-CAA CTT CAT CCA CGT TCA CC-3′, 110bp) served as the positive control for successful DNA purification. The tumor samples were examined for HPV infection using two-step PCR. General HPV DNA presence was determined using the degenerated primers MY09 (5′-CGT CCM ARR GGA WAC TGA TC-3′) and MY11 (5′-GCM CAG GGW CAT AAY AAT GG-3′), which were used to amplify a 450 bp fragment of the viral L1 gene.

The second step included typing of the HPV-positive samples for HPV-16 and HPV-18 using specific primers. For HPV-16 detection, the primer set 5′-GTC AAA AGC CAC TGT GTC CT-3′ and 5′-CCA TCC ATT ACA TCC CGT AC-3′ amplified a 499 bp fragment within the E7, E6, and E1 genes. HPV-18-specific primers (5′-CCG AGC ACG ACA GGA ACG ACT-3′ and 5′-TCG TTT TCT TCC TCT GAG TCG CTT-3′) attached in the E6 and E7 gene region and generated a PCR product of 172 bp. All primers were customized by Invitrogen (Darmstadt, Germany). For validation, one HPV-16 and one HPV-18-positive sample were used as positive controls. PCR was performed with Platinum Blue PCR SuperMix (Invitrogen). For amplification, 40 ng of purified DNA and 200 nM of each primer were added and a hot-start PCR was accomplished using a Mastercycler (Eppendorf, Hamburg, Germany). After preheating at 95 °C, the PCR samples were processed for 30 s at 95 °C, 30 s at 50 °C, and 60 s at 72 °C for 45 cycles. The DNA fragments were separated on 2% agarose gel in 0.5× Tris/Borate/EDTA (TBE) buffer at 150 V. The gel was stained with 3× GelRed (Biotrend, Cologne, Germany) diluted in water and the bands were analyzed with a Kodak Image Station 440CF (Boston, MA, USA). As a method of quality control, HPV DNA sequencing was conducted. Nested PCR was performed first using an MY09/11 primer set, and GP5 (5′-TTT gTT ACT gTg gTA gAT ACY AC-3′) and GP6 (5′-GAA AAA TAA ACT GTA AAT CAT ATT C-3′) primers afterward (annealing at 40 °C) (Remmerbach et al., 2004). Then, the amplicon (<140 bp) was sequenced.

#### 2.4.3. p16 Immunohistochemistry

Paraffin or cryosections of tumor tissue were stained for p16 using the CINtec Histology Kit (Roche, Mannheim, Germany), according to the manufacturer’s instructions. DAB staining for the expression of p16 was classified by the amount of positive tumor cells (threshold 75%), the localization (nuclear, cytoplasmatic), and the staining intensity, i.e., none (−), low (+), medium (++), or high (+++).

### 2.5. Cell Lines

Detroit-562 (CCL-138; American Type Culture Collection), PE/CA-PJ15 (European Collection of Authenticated Cell Cultures), and Cal-27 (German Collection of Microorganisms and Cell Cultures) cell lines were maintained in a tumor culture medium (Tumor Plus 263, Lot CP16-1426, Capricorn, Ebsdorfergrund, Germany) at 37 °C and 5% CO2 Gentamicin sulfate at a concentration of 0.1 mg/mL (17-518Z, Lonza, Basel, Switzerland) was added to the medium. Cell detachment was done by incubating cells for 13 minutes with Accutase (A6964; Sigma, St. Louis, CA, USA). Cell counting was performed with the cell counting system CedexXS (Roche Diagnostics GmbH, Rotkreuz, Switzerland). Cells were passaged weekly.

### 2.6. Migration Assay and Cell Culture

For migration testing, cells (Detroit-562: 5000; Cal-27: 4000; PE/CA-PJ15: 3900) were cultured on a Culture-Insert 3 Well in a 35 mm μ-Dish (80366; Ibidi, Martinsried, Germany) until confluent. Cells were then treated with 10 μg/mL Mitomycin-C (BML-GR311-0002; Enzo, Lörrach, Germany) for two hours. After removing the silicone gasket, cells were washed with DPBS and then incubated with a fresh pure medium, 40 ng/mL NGF (450-01, PeproTech), 40 ng/mL BDNF (248-BD-005, BioTechne, Minneapolis, MN, USA), 40 ng/mL NT3 (450-03, PeproTech, Rocky Hill, CT, USA), 4 μM CD271 inhibitor LM11A 31 dihydrochloride (21982; Cayman, Ann Arbor, MI, USA), and 4 μM Trk Inhibitor PF-06273340 (PZ0254; Sigma). The incubation times varied dependent on proliferation rates: PE/CA-PJ-15—7h; Cal-27—19 h; and Det 562—24 h. Gap size was quantified by the number of pixels measured with Adobe Photoshop Elements 13 version 13.0 (Adobe Inc., San José, CA, USA).

To investigate proliferation depending on the stimulation or inhibition of neurotrophin receptors, 10,000 cells of each cell line were cultured with a pure tumor culture medium and with NGF, BDNF, and NT3 as additional neurotrophins in different concentrations, i.e., 10, 20, or 40 ng/mL. Also, the CD271 inhibitor LMA11A 31 dihydrochloride and the Trk inhibitor PF-06273340 were tested. Cells were counted after seven days.

### 2.7. Fluorescence-Activated Cell Sorting (FACS)

Anti-LNGFR PE (Anti-Low-affinity nerve growth factor receptor CD271 PE, mouse, 130-098-111; Miltenyi, Bergisch Gladbach, Germany), Anti-TrkA AlexaFluor 488 nm (FAB1751G; R&D, Minneapolis, MN, USA), Anti-TrkB AlexaFluor 405 nm (FAB3971V-100UG; R&D, Minneapolis, MN, USA), Anti-TrkC AlexaFluor 647 nm (FAB3731R; R&D, Minneapolis, MN, USA), and anti-mouse-IgG1 PE (12-4714-41, ThermoFischer) were used. The cells were incubated for 20 minutes at 4 °C. Cell sorting was done by FACSAriaFusion (BD, Franklin Lakes, NJ, USA). Two thousand cells were cultured in 6-well (657160; Greiner, Frickenhausen, Germany) and 96-well plates (655180; Greiner, Frickenhausen, Germany). Cell colonies were counted after seven days. The cell-doubling time was calculated using the following formula: t_doubling_ [days] = log_e_ (2)/[(log_e_ (cell number end) − log_e_ (cell number start))/ incubation time [days])]. Cells were sorted according to four expression patterns: CD271^negative/low^, CD271^low^, CD271^medium^, and CD271^high^ cells. For the single cell assay, single CD271^high^ cells (highest 2% of CD271-positive cells) were segregated, sorted into 96-well plates, and incubated for 7 days. Finally, the number of colonies was counted.

### 2.8. Statistical Analysis

Statistical analysis was done using SPSS Statistics version 24.0.0.0 (IBM Corporation, NYC, NY, USA). All experiments were performed at least 3 times. Cell-doubling times and the number of colonies were compared using one-way ANOVA, including the Bonferroni post hoc test or nonparametric tests (Kruskal–Wallis test). Differences between two groups were analyzed using the unpaired t-test. Patients’ data were analyzed in total and based on their expression profiles of CD271, TrkA, TrkB, and TrkC according to primary tumor location, TNM stage, G stage, UICC stage, alcohol and tobacco consumption, patient age, progression, disease-free survival (DFS), overall survival (OS), p16/18 status, keratinization, and therapy (surgery, radiochemotherapy (RCT), surgery + RCT). The frequency of different factors, such as metastases, depending on expression of neurotrophic receptors was compared using cross-supplemental tables (Pearson–chi^2^ test). The Kaplan–Meier log-rank test was used to test the dependence of survival rates on different factors. All analyses were performed on the basis of bilateral distribution. *p* values of <0.05 were considered to be statistically significant.

## 3. Results

### 3.1. Expression of CD271, TrkA, TrkB, and TrkC in Human Primary HNSCC

The expression profiles of CD271 were very heterogeneous, whereas the staining intensity among the cells showed no or only very slight differences. Following this observation, carcinomas were classified by the amount of CD271+ cells (Figure 1 and Table 2). More than 10% of cells were CD271+ in the majority of the tumors (113/184; 61%). These tumors were considered to be CD271+. However, between the different carcinomas, the cell count of CD271+ tumor cells varied from a low percentage to a majority of tumor cells (Figure 1C). In carcinomas with heterogeneous CD271 expression, the CD271+ cells were located at the border of the cell nests and in the invasive front. CD271− tumors, of which CD271+ cells made up less than ten percent of the total, were in the minority (71/184; 39%); in these cases, CD271+ cells were found alone in the center of the tumor nests (Figure 1A). More than half of the CD271− tumors were located in the larynx (37/71; 52%). Compared to the number of CD271− tumors in hypopharyngeal (14/71; 20%) and oropharyngeal regions (20/71; 28%), CD271− tumors occurred significantly more frequently in the larynx (*p* = 0.043; Pearson R = 0.185; Pearson–chi^2^ test).

TrkA expression was more homogeneous than CD271 expression. Most samples were positive for TrkA (149/184; 81%). TrkA-positive cells were found throughout the whole tumor cell nest, including the margin and the center (Figure 1D). Most of CD271+ tumors overlapped with TrkA+ expression (98/113; 87%), but 40% (6/15) of CD271+/TrkA– tumors were considered to be HPV-mediated. In contrast, only 7% of CD271+/TrkA+ carcinomas were referred to HPV (*p* < 0.001; Pearson R = 0.349; Pearson–chi^2^ test).

Fewer samples were positive for TrkB (102/184; 55%). TrkB and CD271 expression often intersected (64/113; 57%), and both were represented more marginally within the tumor cell nests. (Figure 1E,H).

Overall, TrkC immunohistochemistry exposed the least number of positive tumor samples (83/184; 45%). Noticeably, the majority of TrkC^+^ cells were represented in the middle of the tumor cell nests contrary to CD271 and TrkB expression (Figure 1E,I). TrkC expression was present in half of the CD271+ tumors (58/113; 51%). The staining patterns of CD271 and TrkC partly overlapped.

### 3.2. High Expression of CD271 Correlates with Reduced Overall Survival and an Increased Number of Distant Metastases

The cumulative overall survival within five years after diagnosis (OS) was compared between the different groups of CD271− expressing tumors. Patients with CD271-expressing carcinomas showed significantly worse survival rates compared to patients with CD271− carcinomas (*p* = 0.009; log-rank test) (Figure 2A and Table 2). CD271+ primary tumors showed a significantly higher presence of distant metastases compared to CD271− tumors (*p* = 0.012; Pearson R = 0.211; Pearson–chi^2^ test) (Table 2). Tumors that expressed CD271+ and TrkC+ led to lower rate of mortality among patients at the end of the investigated five year period than CD271+/TrkC− cases, even though this difference was not statistically significant (Figure 2B). Independent from CD271 expression, patients with HPV-mediated carcinomas had better OS (*p* = 0.009; log-rank test) compared to cases to HPV.

### 3.3. Relevance of CD271 Expression in Response to Treatment

In general, patients were treated with surgery, radiotherapy/radiochemotherapy (R(C)T), or a combination of both. A priori, carcinomas that received only surgery had better prognoses due to their low UICC stages. In order to find out more about the prognostic value of CD271, we investigated the overall survival of patients after receiving any treatment relating to their CD271 expression. CD271+ tumors that were treated with single R(C)T were mostly UICC4, which may explain the low survival rate. However, CD271− tumors treated with R(C)T showed an overall survival rate of 100%. Due to the low number of patients in this group (n = 2), the result may not be representative, but it is statistically significant (*p* = 0.014; log-rank test). The survival rates of both surgery groups did not differ. The comparison between the groups of tumors that were treated with surgery and R(C)T showed a better overall survival rate in the CD271− group than in CD271+ group.

### 3.4. TrkC/CD271 Expression is Associated with Less Distant Metastases

The role of TrkC/CD271 expression in human primary HNSCC was evaluated. Remarkably, 36% (14/39) of CD271+/TrkC− carcinomas showed distant metastases, whereas carcinomas with expression of both receptors showed metastases in only 16% (7/45) of cases (Table 2). To sum up, TrkC- and CD271-expressing tumors showed a significantly reduced association with distant metastases compared to CD271+/TrkC− carcinomas (*p* = 0.032; Pearson R = 0.234; Pearson–chi^2^ test). Another remarkable observation was that TrkC+ tumors were significantly linked to keratinization in human primary HNSCC (*p* < 0.001; Pearson R = 0.316; Pearson–chi^2^ test).

### 3.5. Expression of CD271 is Associated with High Cell Replication

In the cell lines, the flowcytometric analysis exposed that nearly all tumor cells showed at least a low level of CD271 expression. In order to find out how the different levels of CD271 expression differed from each other, staining intensity was subdivided into four groups: negative/very low, low, medium, and high (Figure 3C, Appendix A). Tumor cells of each group were sorted by flow cytometry (FACS). After seven days of incubation, the colony numbers were counted (Figure 3B). CD271^high^ cells of all three cell lines showed a significantly higher number of colonies compared to CD271^very low/negative^ cells (*p* < 0.001; t-test). In each cell line, CD271^high^ and CD271^medium^ cells showed significantly faster cell-doubling times compared to CD271^very low/negative^ cells. Furthermore, Detroit-562 and Cal-27 cell lines showed significant differences between the CD271^high^ or CD271^medium^ groups compared to the CD271^low^ group. Even CD271^low^ cells of Detroit-562 and Cal-27 cell lines showed significantly reduced replication times compared to CD271^very low/negative^ cells. In all cell lines, there were no differences between CD271^high^ and CD271^medium^ cells. This result implies that CD271 expression correlates with proliferative characteristics of human HNSCC cell lines, but cannot be increased further if a certain threshold of CD271 expression is reached.

Cells of each population were reseeded and incubated for another week. There were no more differences in the cell-doubling times at all. In addition, negative cells were able to regenerate populations, including both positive and negative cells. Immunohistochemical staining showed no more differences between the groups (Appendix A). This result implies that heterogeneous populations, including CD271+ and CD271– cells can be formed from former CD271^high^ cells and CD271^high^ cells can initiate heterogeneous cell populations.

Moreover, single CD271^high^ cells were segregated and sorted into 96-well plates using flow cytometry. After seven days of incubation, colonies were counted. The CD271^high^ cells of all three cell lines showed very high colony formation ability: 41% ± 3.8% (CA/PE-PJ15), 36.1% ± 3.1% (Cal-27), and 29.2% ± 3.6% (Detroit-562). Only 6% ± 0.9% (CA/PE-PJ15), 4.7% ± 0.4% (Cal-27), and 2.3% ± 0.4% (Detroit-562) of CD271^low^ cells were able to initiate a colony.

### 3.6. TrkC Co-Expression Reduces Proliferation in CD271^high^ Cells

The proliferative properties of the CD271^high^ cells were analyzed in vitro regarding their Trk co-expression. All cell lines showed co-expression of CD271 and TrkC. The tumor cells of each cell line were sorted according to four expression patterns: CD271+/TrkC+, CD271−/TrkC+, CD271+/TrkC−, and CD271−/TrkC− (Figure 4B and Appendix A). After two weeks of incubation, cell numbers were counted and generation times were calculated. In the Detroit-562 and Cal-27 cell lines, CD271−/TrkC+ cells showed significantly decreased cell-doubling times compared to any other population. In addition, the cell-doubling times of CD271+/TrkC− cells were significantly higher compared to CD271+/TrkC+ and CD271−/TrkC+ cells. In PE/CA-PJ15 cells, there were no differences between the populations. Co-expression patterns of CD271/TrkA were also evaluated by flow cytometry, but the cell generation times did not differ. None of the cell lines expressed TrkB.

### 3.7. Dependence of Cell Motility and Mitotic Rates on Stimulation and Inhibition of Neurotrophic Receptors

To investigate the effect of the stimulation or inhibition of CD271 and Trks on cell motility in vitro, we performed migration assays with cells that were treated differently compared to control cells incubated in a pure tumor medium. Neurotrophic factors NGF, BDNF, and NT3 (40 ng/mL for each factor), as well as CD271 inhibitor LM11A 31 dihydrochloride (40 ng/mL), Trk inhibitor PF-06273340 (40 ng/mL), or the combination of LM11A 31 and PF-06273340, were added. After seven hours of incubation, the unmigrated area was measured. PE/CA-PJ15 cells treated with LMA11A 31 showed significantly reduced motility compared to the control cells (Figure 5), while neither neurotrophic factors nor the Trk inhibitor PF-06273340 affected cell motility in vitro. Inhibition by LMA11A 31 was independent of incubation with neurotrophic factors or the combination with PF-06273340. However, neither Cal-27 nor Detroit-562 cells were significantly affected by any agent, including LMA11A 31 (Appendix A).

To investigate the mitotic rates for their dependence on the stimulation or inhibition of neurotrophic receptors, adherent tumor cells were treated with NGF, BDNF, NT3, LMA11A 31, and PF-06273340. However, no agent had a significant impact on cell proliferation.

## 4. Discussion

In this study, we undertook a retrospective analysis of human HNSCC to investigate the role of neurotrophin receptors in HNSCC. We found that CD271 was heterogeneously expressed in the majority of human HNSCC, which was consistent with the observations of previous studies [5,12,18]. Patients with CD271− tumors had a better response to treatment. In our study, the group of R(C)T-treated (radiotherapy or radiochemotherapy) CD271− patients was very low. In order to further determine whether radio(chemo)therapy is beneficial in CD271− patients, further studies must be undertaken. The majority of CD271− carcinomas were located in the larynx. Tumors occurring in this region were considered to have a more favorable prognosis than HNSCCs in other locations [23].

It is also notable that CD271+ cells were found to be more marginally within the cell nests with respect to the invasive front. The high expression of CD271 was related to the presence of distant metastases and may be one reason that this type had a lower rate of survival. So far, correlations between the number of CD271+ cells and clinical outcome, response to treatment, and distant metastases have been reported for esophageal cancer and melanoma [16,24,25]. Thus, these results suggest that the number of CD271+ cells provides prognostic information in patients with HNSCC. This might include the overall survival probability and the tumor’s response to radio(chemo)therapy, respectively.

The vast majority of tumors expressed TrkA, while TrkB and TrkC were expressed in only half of the tumor samples. These quantities of receptor-positive cells match the findings by Sasahira et al. [26], but the amount of TrkB+ cells varied from 30% to 70% between different studies [27,28,29]. With the exception of a correlation between TrkC expression and keratinization, we could not find any further relationship between Trk receptor expression and clinical data, including tumor localization, HPV infection, survival, tumor stage, or metastases. CD271+ tumors were also considered with regard to their Trk expression patterns, because neurotrophins also bind to Trk receptors; earlier studies indicated that even the functions of CD271 can vary depending on Trk expression [7,9,21,22]. Interestingly, CD271+ tumors lacking TrkA expression were mostly HPV-mediated. A study by Dudas et al. [30] investigated TrkA and CD271 expression related to HPV infection, but the proportions of CD271+ and TrkA+ cells did not differ according to HPV infection. A further remarkable observation was that CD271+ tumors lacking expression of TrkC showed a higher correlation with distant metastases and lower survival rates compared to tumors with expression of both CD271 and TrkC. As mentioned above, TrkC expression is associated with tumor keratinization. Since keratinization represents more tumor differentiation and is a favorable prognostic factor for HNSCC [31], this could be one explanation for our observations, even though we could not find a direct association between keratinization and clinical outcome. Moreover, high TrkC expression as a predictor for favorable clinical outcomes was also observed in breast cancer [32].

In this study, we used in vitro experiments to substantiate the results of the retrospective study and to further investigate the function of CD271 and the Trk family in terms of proliferation and migration. Using flow cytometry, we found that mitotic rates gradually increased along with CD271 expression. CD271 expression as a marker for higher cell proliferation was also reported in several studies [5,12,18], but it was only distinguished between CD271+ and CD271− cells; different degrees of CD271 expression were not considered. However, mitotic rates were unaffected by the stimulation (neurotrophins) or inhibition (LM11A 31) of CD271, suggesting that CD271 only indirectly serves as a surface marker for a highly proliferative subpopulation. In addition, co-expression of CD271 and Trk receptors was investigated using flow cytometry; these receptors were sorted according to different co-expression patterns. All tested cell lines were negative for TrkB. Mitotic rates of CD271+ were not affected by TrkA co-expression, but were affected by TrkC co-expression. CD271/TrkC-co-expressing cells showed significantly decreased mitotic rates compared to CD271+ cells lacking TrkC. This observation suggests that a highly proliferative CD271+ subpopulation can be further narrowed by considering TrkC co-expression. The co-expression of CD271 and TrkC led to reduced mitotic rates in vitro, which suggests that this might help to narrow the proliferative CD271+ subpopulation. Moreover, TrkC expression enhances the prognosis of human CD271+ tumors, probably because it is associated with fewer distant metastases. TrkC also augments tumor differentiation by activating MAPK independent of CD271, which in turn results in a less aggressive phenotype [32,33,34]. This is consistent with our observation that TrkC is associated with keratinization and a more differentiated cancer phenotype. We saw no sign of the CD271 and TrkC receptors working together.

Lastly, we examined whether cell motility was affected by the stimulation (neurotrophins) or inhibition (CD271 inhibitor LM11A 31; Trk Inhibitor PF-06273340) of CD271 and Trk receptors. We found that the inhibition of CD271 by the small molecule ligand LMA11A 31 caused a decrease in cell motility of PE/CA-PJ15 cells. However, Cal-27 and Detroit-562 cells were unaffected by LMA11A 31, implying that the inhibition of CD271 may reduce cell motility directly, but not in every tumor. The Trk inhibitor PF-06273340 had no impact on in vitro cell migration, not even in combination with the CD271 inhibitor LM11A 31. This indicated that CD271 works alone and not in conjunction with Trk receptors to effect migration in PE/CA-PJ15 cells. Consistent with our findings, antibody targeting of CD271 sufficiently suppressed melanoma metastases in patient-derived xenografts [35]. In different cancers, activation of RhoA has been shown to promote cell migration and invasion by reorganization of the cytoskeleton [36,37,38]. Several studies have shown that CD271 activation promotes RhoA activity in HNSCC [12,18]. Other studies suggested endothelial cell-specific molecule 1 (ESM1) to be a downstream target of CD271 signaling pathways because knockdown of ESM1 counteracted metastatic properties in CD271-overexpressing HNSCC cells [14]. However, this reduction in cell migration was only seen in one of three cell lines. This may indicate differences between HNSCCs, which need to be further investigated.

The expression of CD271 predicts the clinical outcome of patients with HNSCC and confers a highly proliferative and resilient subpopulation. The inhibition of CD271 may be a promising therapeutic target to reduce distant metastases and improve prognoses.

However, proliferation was unaffected by CD271 or Trks inhibition in vitro. In the future, the role of neurotrophin receptors in proliferation must be further investigated. Moreover, the impact of TrkB, as well as the relationship between TrkA and HPV infection in HNSCC, should be examined.

## Figures and Tables

**Figure 1 cells-08-01167-f001:**
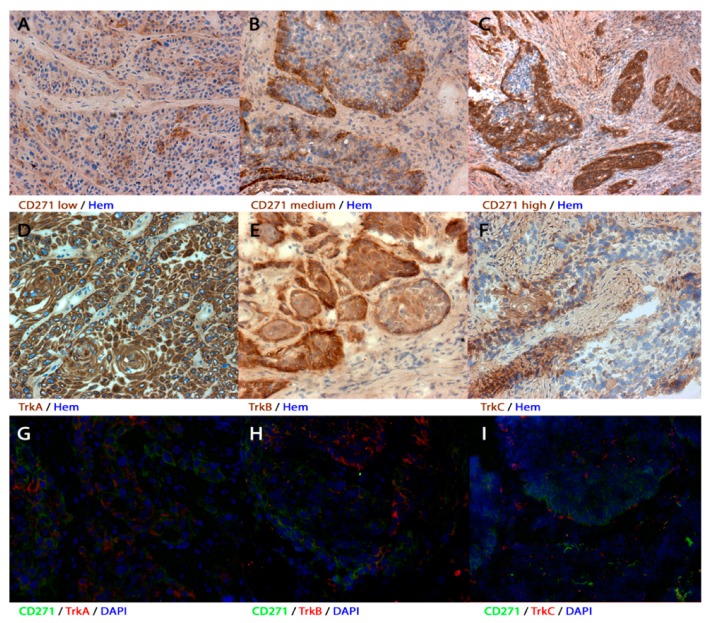
Expression patterns of the receptors in human primary head and neck squamous cell carcinoma (HNSCC) (**A**–**F**) immunohistochemistry and (**G**–**I**) immunofluorescence of human primary tumors. (**A**–**C**) Tumor samples showed different amounts of CD271+ cells. (**A**) If <10% of the cells were stained, the tumor was considered to be receptor-negative. (**B**,**C**) Tumor samples with >10% stained cells were considered to be receptor-positive. The amount of positive cells varied from just over 10% (**B**) to the entire cell nest (**C**). (**D**–**F**) Representative tumor sections with TrkA+, TrkB+, or TrkC+ cells. TrkA was mostly expressed in the majority of the cells, whereas TrkB was represented rather marginally. Most of the samples only expressed TrkC in very few cells. (**G**–**I**) CD271 and TrkA/B/C expression patterns partly overlapped. Immunohistochemical sections were counterstained with hematoxylin (hem) and immunofluorescent sections were counterstained with 4′6-Diamidin-2-phenylindol (DAPI). Magnification (**A**,**B**,**D**–**H**) ×200; magnification (**C**,**I**) ×100.

**Figure 2 cells-08-01167-f002:**
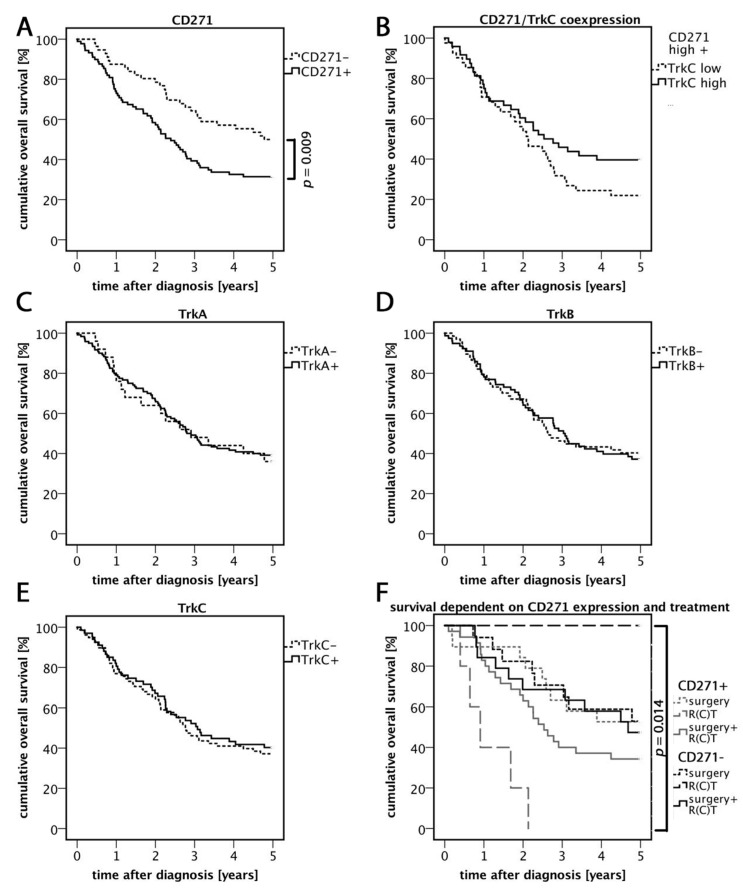
Five year overall survival rates. (**A**) Cumulative overall survival five years after diagnosis related to CD271 expression profiles. CD271+ carcinomas showed significantly reduced overall survival rates compared to CD271− tumors. (**B**) Overall survival dependent on CD271/TrkC expression. CD271+/TrkC+ carcinomas showed higher survival rates compared to CD271+/TrkC−, however this difference was not significant (*p* = 0.128; log-rank test). (**C**–**E**) Survival rates dependent on Trk family member expression. (**F**) Overall survival dependent on CD271 expression and therapy. The best survival rate was demonstrated in CD271− carcinomas that received R(C)T. The worst survival rate was in CD271+ carcinomas that received R(C)T. This difference was significant.

**Figure 3 cells-08-01167-f003:**
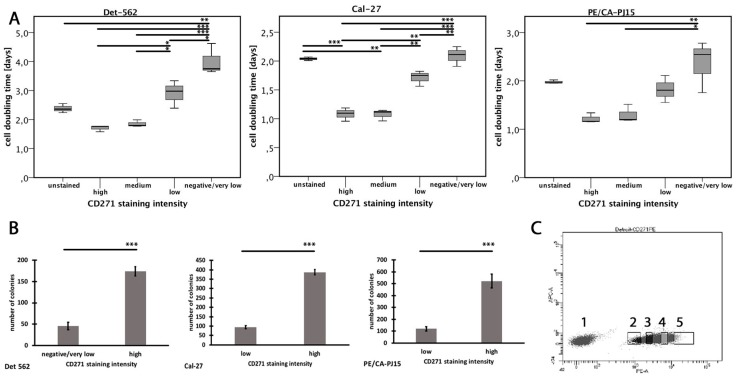
Cell proliferation dependent on CD271 expression. (**A**) Cell-doubling times of different CD271 staining intensities after 14 days of incubating (*p* < 0.05*, *p* < 0.01**, *p* < 0.001***; PE/CA-PJ15: Kruskal–Wallis test; Det-562, Cal-27: ANOVA; for exact *p*-values, see Appendix A). (**B**) Number of colonies after 7 days of incubation. (**C**) Exemplary staining of the Detroit-562 cell line. Cells were sorted into CD271^negative/low^ (2), CD271^low^ (3), CD271^medium^ (4), and CD271^high^ cells (5). (*p* < 0.001***; t-test).

**Figure 4 cells-08-01167-f004:**
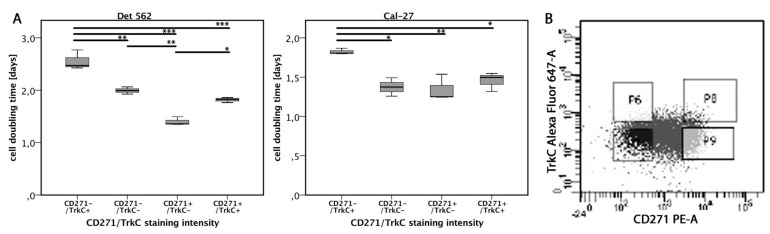
Cell proliferation dependent on CD271/TrkC co-expression. (**A**) Cell-doubling times of cells with different CD271/TrkC co-expression patterns. TrkC+/CD271+ means high expression of CD271 and high co-expression of TrkC. TrkC–/CD271– means low expression of CD271 and low co-expression of TrkC, etc. (**B**) Exemplary TrkC/CD271 staining pattern of Detroit-562 cell line. The cells of each cell line were sorted according to the following gates. Upper right: CD271+/TrkC+; upper left: CD271−/TrkC+; bottom right: CD271+/TrkC−; bottom left: CD271−/TrkC− (*p* < 0.05*, *p* < 0.01**, *p* < 0.001***, ANOVA; for exact *p*-values, see Appendix A).

**Figure 5 cells-08-01167-f005:**
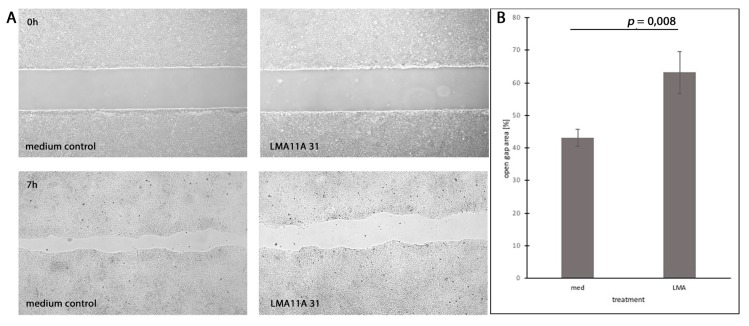
Inhibition of cell motility by CD271 ligand LMA11A 31. (**A**) PE/CA-PJ15 cells treated with LMA11A 31 hydrochloride (LMA) compared to the medium control group (med). (Top) Gap area just after removing the silicone gasket. (Bottom) Gap area after 7 hours of incubation. (**B**) Quantification of open gap area after 7 h related to the initial area. (*p* = 0.008; t-test).

**Table 1 cells-08-01167-t001:** Patient data. Five-year overall survival (OS) describes the survival rate within five years after diagnosis. Five-year disease-free survival (DFS) describes the length of time without any disease symptoms. Tumor size (T stage), lymph node metastasis (N stage), distant metastasis (M stage), grading (G stage), and UICC (Union Internationale Contre le Cancer) staging were done in accordance with the 7th edition of the AJCC (American Joint Committee on Cancer) cancer staging manual. Patients were treated with surgery, single radio(chemo)therapy (R(C)T; (radiochemotherapy or radiotherapy), or a combination of both (surgery + R(C)T). Therapy decisions were made depending on the UICC stage and patients’ comorbidities. Carcinomas exhibiting keratin pearls were classified as keratinized. Also human papillomavirus (HPV) mediation was considered (HPV-Mediated).

Patient Data	Cases	[%]
**Gender**	184	100.0
male	161	87.5
female	23	12.5
**Age (years)**	184	100.0
median ± standard error	57.09 ± 9.027	
youngest	29	
oldest	79	
**5 year OS (days)**	163	88.6
median ± standard error	1137 ± 647	
**5 year DFS (days)**	56	30.4
median ± standard error	720 ± 623	
**Tumor Location**	184	100.0
oropharynx	54	29.3
hypopharynx	53	28.8
larynx	77	41.8
**T stage**	158	85.9
T1	25	15.8
T2	69	43.7
T3	32	20.3
T4	32	20.3
**N stage**	174	94.6
N0	70	40.2
N1	25	14.4
N2	63	36.2
N3	16	9.2
**M stage**	143	77.7
M0	117	81.8
M1	26	18.2
**UICC stage**	174	94.6
UICC 0	1	0.6
UICC 1	17	9.8
UICC 2	33	19.0
UICC 3	27	15.5
UICC 4	96	55.2
**G stage**	172	93.5
G1	12	7.0
G2	125	72.7
G3	35	20.3
**Alcohol and Tobacco Consumption**	106	57.6
none	14	13.2
tobacco	16	15.1
alcohol	8	7.5
tobacco + alcohol	68	64.2
**Therapy**	120	65.2
surgery	47	39.2
R(C)T	8	6.7
surgery + R(C)T	65	54.2
**HPV-Mediated**	19	10.3
HPV 16	18	94.7
HPV 18	1	5.3
**Keratinization**	184	100.0
no	107	41.8
yes	77	58.2

**Table 2 cells-08-01167-t002:** Expression patterns in human primary HNSCC according to tumor localization, HPV mediation, M Stage and 5 year overall survival (5 year OS). Out of 63 CD271+ tumors, 21 showed distant metastases (25%). Compared to 8.5% (5/59) of CD271– carcinomas, this difference was statistically significant (I; *p* = 0.012; Pearson R = 0.211; Pearson–chi^2^ test). Patients with CD271+ carcinomas had a significantly reduced 5 year OS compared to patients with CD271– carcinomas (II; *p* = 0.009; log-rank test). CD271− tumors were more frequent in the larynx than in the oropharynx or hypopharynx (III; *p* = 0.043; Pearson R = 0.185; Pearson–chi^2^ test). Forty percent (6/15) of CD271+/TrkA– tumors but only 7.1% (7/98) of CD271+/TrkA+ tumors were considered to be HPV-mediated (IV; *p* < 0.001; Pearson R = 0.349; Pearson–chi^2^ test), and 25.4% (14/55) of CD271+/TrkC– tumors but only 12% (7/58) of CD271+/TrkC+ tumors were related to distant metastases (V; *p* = 0.032; Pearson R = 0.234; Pearson–chi^2^ test). Patients with CD271+/TrkC+ tumors showed worse survival compared with CD271+/TrkC– patients, however this difference was not significant (VI; *p* = 0.128; log-rank test).

Expression	Overall		Oropharynx		Hypopharynx		Larynx		HPV-mediated		M0		M1		5 year OS (days)
	**Cases**	**(%)**	**Cases**	**(%)**	**Cases**	**(%)**	**Cases**	**(%)**	**Cases**	**(%)**	**Cases**	**(%)**	**Cases**	**(%)**	**Median ± standard error**
**CD271+**	113	61	34	18	39	21	40	22	13	12	63	44	21^I^	15	1024 ± 64^II^
**CD271−**	71	39	20	11	14	8	37^III^	20	6	8	54	38	5	3	1325 ± 77
**TrkA+**	149	81	49	27	41	22	59	32	13	9	96	67	21	15	1137 ± 55
**TrkA−**	35	19	5	3	12	7	18	10	6	17	21	15	5	3	1136 ± 127
**TrkB+**	102	55	27	15	32	17	43	23	11	11	67	47	11	8	1136 ± 68
**TrkB−**	82	45	27	15	21	11	34	18	8	10	50	35	15	10	1137 ± 77
**TrkC+**	83	45	24	13	24	13	35	19	6	7	59	41	8	6	1174 ± 76
**TrkC−**	101	55	30	16	29	16	42	23	13	13	58	41	18	13	1106 ± 69
**CD271+/TrkA+**	98	87	32	28	32	28	34	30	7	7	54	64	19	23	1014 ± 69
**CD271+/TrkA−**	15	13	2	2	7	6	6	3	6^IV^	40	9	11	2	2	1103 ± 185
**CD271+/TrkB+**	64	57	17	15	25	22	22	12	6	9	36	43	10	12	969 ± 88
**CD271+/TrkB−**	49	43	17	15	14	12	18	10	7	14	27	32	11	13	1091 ± 95
**CD271+/TrkC+**	58	51	16	14	20	18	22	12	5	9	38	45	7^V^	8	1101 ± 94^VI^
**CD271+/TrkC–**	55	49	18	16	19	17	18	10	8	15	25	30	14	17	944 ± 88

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
