# Peer review of "Relevance of Neurotrophin Receptors CD271 and TrkC for Prognosis, Migration, and Proliferation in Head and Neck Squamous Cell Carcinoma"

_cells, 2019, doi:10.3390/cells8101167_

Round 1

Reviewer 1 Report

The manuscript entitled “The Role of Nerve Growth Factor Receptors CD271, TrkA, TrkB and TrkC in Squamous Cell Carcinoma of the Oropharynx, Hypopharynx and Larynx” by Yannick Foerster et al, investigated the role of nerve growth factor receptor CD271 as a candidate for CSC identification and the tropomyosin receptor kinase family (Trk) in HNSCC. The authors showed that these receptors were heterogeneously expressed in human primary HNSCC tissues by IHC. A high percentage of CD271+ cells predicts a worse clinical outcome of HNSCC patients, while patients co-expressed with TrkC showed favorable outcome. In vitro, cells with CD271 expression exhibited higher proliferation rates and the CD271 inhibitor LM11A 31 suppressed cell motility. Coexpression of TrkC was able to counteract the proliferative effect of high CD271 expression in vitro. This study is interesting to the research field, however, the deficiencies in data presentation and interpretation make this manuscript unacceptable in the present form.  The reviewer suggests a substantial revision of the manuscript, so it will meet standards of publications in this peer-reviewed journal. The specific comments are as follows: 

The title should present a major conclusion of the paper. However, from the current title, the readers cannot get any information about what kind roles of CD271 and TrkA/B/C in HNSCC. Introduction did not present enough background information. The authors described CD271 is a low affinity NGFR, while TrkA/B/C are the high affinity receptors. Do those molecules have the DNA or protein sequencing homologies? Are these molecules located in which chromosomes, are in a gene family? What kind known differences in their biological function or downstream signaling? These background information are important to interpret and understand the data from this study. For Table 1, the author should present more detailed figure legends, to explain T, N, M, UICC, and G stages, as well as noxes for general readers. In Fig 1 and Table 2, the authors presented comparable IHC and IF staining of CD271, and TrkA/B/C. However it is not very clear why the authors only focus the following studies on CD271 and TrkC, but not the other two molecules TrkA/B. Please explain. Please add the statistical analysis to Table 2, and analyze the expression correlations among the 4 molecules. Please analyze the expression levels of the 4 molecules in the tissues with different HPV status and primary tumor locations. Please show the patient’s survival data with TrkA/B/C IHC staining scores. Any statistical significance? Please explain why high expression of CD271 related to a worse patient outcome, while high expression of TrkC showed a better survival? Please show the patient survival curves based on the two molecules IHC staining, as the single positive, double positive and double negative with statistical analysis. The labels of Fig 3 and 4 are too small, which are difficult for the reviewer to judge the results presented. Fig 5 only presented data from one cell line. It will be nice to present data from a second cell line. Discussion needs to be revised to better interpret the results, the underline scientific mechanisms, and the clinical significance.

Author Response

Point 1: The title should present a major conclusion of the paper. However, from the current title, the readers cannot get any information about what kind roles of CD271 and TrkA/B/C in HNSCC.

Response 1: We changed our title into "Relevance of Neurotrophin Receptors CD271 and TrkC for prognosis, migration and proliferation in Head and Neck Squamous Cell Carcinoma" in order to present our major conclusions

Point 2: Introduction did not present enough background information. The authors described CD271 is a low affinity NGFR, while TrkA/B/C are the high affinity receptors. Do those molecules have the DNA or protein sequencing homologies? Are these molecules located in which chromosomes, are in a gene family? What kind known differences in their biological function or downstream signaling?

Response 2: We gave a few examples why it is important to consider Trk coexpression when investigating CD271 

Point 3: For Table 1, the author should present more detailed figure legends, to explain T, N, M, UICC, and G stages, as well as noxes for general readers

Response 3: Explanation of every abbreviation is now indicated in our Material & Methods section

Point 4: However it is not very clear why the authors only focus the following studies on CD271 and TrkC, but not the other two molecules TrkA/B. Please explain.

Response 4: We investigated each Trk receptor, especially in combination with CD271. However, we couldn't find any significance between the other Trk receptors and clinical data indicated in Table 1. We readjusted both our introduction and discussion to explain why we focused our analysis on TrkC.

Point 5: Please add the statistical analysis to Table 2, and analyze the expression correlations among the 4 molecules. Please analyze the expression levels of the 4 molecules in the tissues with different HPV status and primary tumor locations.

Response 5: Every Trk receptor, alone and in combination with CD271, was analyzed with HPV status and primary tumor locations. An association between CD271+/TrkA- expression and HPV mediation was found.

Point6: Please show the patient’s survival data with TrkA/B/C IHC staining scores. Any statistical significance?

Response 6: Survival data was added to Figure 3. We couldn't find any significance. However, CD271+/TrkC+ show better survival compared with CD271+/TrkC-.

Point 7: Please explain why high expression of CD271 related to a worse patient outcome, while high expression of TrkC showed a better survival? Please show the patient survival curves based on the two molecules IHC staining, as the single positive, double positive and double negative with statistical analysis.

Response 7: We undertook several functional tests and a retrospective analysis to clarify whether CD271 and Trk receptors are related to outcome and higher mitotic rates. Further studies have to investigate the molecular mechanisms. The survival curves dependent on expression of two molecules showed no differences.

Point 8: The labels of Fig 3 and 4 are too small, which are difficult for the reviewer to judge the results presented

Response 8: We enlarged the labels.

Point 9: Fig 5 only presented data from one cell line. It will be nice to present data from a second cell line.

Response 9: We added the other cell lines to the supplemental material.

Point 10:Discussion needs to be revised to better interpret the results, the underline scientific mechanisms, and the clinical significance.

Response 10: The whole discussion has been revised.

Reviewer 2 Report

Abstract

Line 18. The authors claim that overexpression of CD271 is related to bad clinical outcome, then they indicate that coexpression or TrkC was more favorable. From this there are few doubts:

1. Elevated expression or TrkC with CD271 indicated a better prognosis in HNSCC patients?

2. Please define adequately what the authors are referring to with coexpression of TrkC.

Is it possible that the authors include the coexpression of TrkA and TrkB, and their significance in the study?

In the conclusion the authors indicate that CD271 is a promising candidate for target in HNSCC treatment. However, this section is not quite understandable. This could be understood as:

1.In patients with HNSCC CD271+/TrkC- the targeted therapy would be no-functional.

2. Only targeted therapy would work with the co-expression of TrkC in CD271+ HNSCC.

Introduction

The authors explain adequately the characteristics of the family TrkR, but the aim of the study is not quite related with the title. I suggest to modify the title as following: “The Role of Nerve Growth Factor Receptors CD271, TrkR family (A, B and C)…”

I suggest modifying the terms in the title of Oropharynx, Hypopharynx and Larynx, for head and neck.

Material and methods

2.2 Patients:

The entire paragraph must be reviewed and re-write according to the authors’ data, due is not entirely understood the value of 5%.

What were the hypothesis included in the study?

Please review the information and its description regarding clinical data provided.

Table 1. Would it be possible that the authors indicate the rate of age, and the highest and lowest value?

What did the authors decided to use the OS and DFS in days instead weeks?

Why the TNM classification is separated by each value? Would it be possible to use it complete in another table?

Please explain the meaning of the term “noxes” that is used in table 1.

It would be adequate to include meaning of each term in the legend of the figure, in order to understand the terms that were used for the included data.

It is well-known that HPV is a risk factor of HNSCC, here there are two considerations to take in account:

1. How many tumors belong to HPV 16 and to HPV 18?

2. What was the employee methodology to detect the HPV?

3. Would it be possible to include the information of the variables that are aimed to be evaluated with HPV?

Page 3, lines 75-77: It is important to establish adequately the percentages of cell counting. Please review all the rates, due in the value “low” is not well understood what do the authors refer to with very few cells.

Lines 125-126. Are not significative the statistic values under 0.05?

Results

Table 2. Is it necessary to include the term “general expression”? I suggest excluding it as possible.

I suggest that less of 10% of the cell counting to be negative, and >10% could be considered positive.

What do the authors only use M stage? Would it be possible to evaluate the gene expression according to the TNM?

 Page 6, lines 173-176. Please review these lines due they are confusing. Why do the authors are comparing medium and high expression, when in table 2 are comparing negative and positive cases?

Lines 176-178. Would it be possible that these lines could be included in material and methods? It is not necessary to include the numbers of the values when these are included in the table.

Lines 181-182. Why did the authors only decided to evaluate distant metastases instead lymph node metastases?

Table 2. Why do the authors include three values when they are only including positive and negative?

Would it be possible that figure 2B can be divided in B and C?

Why do the authors define the term R(C)T in this section? What did they explain with R(C)T is radiotherapy+ radiotherapy and chemotherapy or separated?

Page 7 line 197. Please specify the meaning of the term “alone”. Are the authors referring to no surgery treatments?

Discussion.

Line 287. Why do the authors make this observation if they did not evaluate cancer stem cells?

Overall, this section is interesting, and it provides novel and valuable data. I consider that the use of abbreviation should be homogenize, and, highlight the importance or TrkA and TrkB. Why do the authors are only focusing in studying specifically TrkC. I suggest that in the discussion they indicate the limitations of the study, and finally conclude the expressions of TrkA and TrkB, due it seems that TrkC unlikely subtypes A and B is directly associated to the prognosis of the patient.

Comments to the authors.

The manuscript is interesting, it provides novel data that could propose new targeted therapies against HNSCC. However, this paper needs a better redaction in the methodology and results. I suggest correcting some mistakes that were included in all the comments.

Author Response

Point 1: The authors claim that overexpression of CD271 is related to bad clinical outcome, then they indicate that coexpression or TrkC was more favorable. From this there are few doubts

Response 1: The meaning of 'better prognosis' was defined more precisely: A high amount of CD271 positive cells predicted a bad clinical outcome of patients with HNSCC and is associated with distant metastases. However, if human carcinoma also expressed TrkC, then this leads to a reduced correlation with distant metastases and better survival rates. In vitro, CD271 expression marked a subpopulation with higher proliferation rates. But the proliferation was lower in tumor cells that coexpressed CD271 and TrkC.

Point 2: Is it possible that the authors include the coexpression of TrkA and TrkB, and their significance in the study?

Response 2: We indicated the meaning of TrkA and TrkB for HNSCC in our Abstract. Furthermore, we explained why TrkC was our main focus.

Point 3: In the conclusion the authors indicate that CD271 is a promising candidate for target in HNSCC treatment. However, this section is not quite understandable. This could be understood as:1.In patients with HNSCC CD271+/TrkC- the targeted therapy would be no-functional.2. Only targeted therapy would work with the co-expression of TrkC in CD271+ HNSCC

Response3: It remains unclear if TrkC coexpression has an impact on the tumor's response to R(C)T. We just observed that expression of both CD271 and TrkC leads to significantly reduced association with distant metastases compared to tumors which express only CD271. Similarly, TrkC coexpression reduced the pro-mitotic effect of CD271 in vitro. The discussion was re-written according to the role of TrkC coexpression. 

Point 4: The authors explain adequately the characteristics of the family TrkR, but the aim of the study is not quite related with the title. I suggest to modify the title as following: “The Role of Nerve Growth Factor Receptors CD271, TrkR family (A, B and C)…” I suggest modifying the terms in the title of Oropharynx, Hypopharynx and Larynx, for head and neck.

Response 4: We adjusted our title in order to present our main conclusions

Point 5: The entire paragraph must be reviewed and re-write according to the authors’ data, due is not entirely understood the value of 5%.What were the hypothesis included in the study?

Response 5: The whole paragraph was we-written according to your comment. Furthermore, we explained our hypotheses in our Introduction

Point6: What did the authors decided to use the OS and DFS in days instead weeks?

Response 6: This led to a finer gradation of the graph.

Point 7: Why the TNM classification is separated by each value? Would it be possible to use it complete in another table? What do the authors only use M stage? Would it be possible to evaluate the gene expression according to the TNM? Lines 181-182. Why did the authors only decided to evaluate distant metastases instead lymph node metastases?

Response 7: This would lead to a very confusing table due to the fact that there are many different combinations of T1/T2/T3/T4/N0/N1/N2/M0/M1. However, every molecule, alone and in combination, was analyzed according to each value. Every significant result is indicated in our text. Furthermore, every value is integrated in the UICC classification.

Point 8: Please explain the meaning of the term “noxes” that is used in table 1.It would be adequate to include meaning of each term in the legend of the figure, in order to understand the terms that were used for the included data

Response 8: We added explanations of every term in our Material & Methods: Patients part.

Point 9: It is well-known that HPV is a risk factor of HNSCC, here there are two considerations to take in account:1. How many tumors belong to HPV 16 and to HPV 18?2. What was the employee methodology to detect the HPV?3. Would it be possible to include the information of the variables that are aimed to be evaluated with HPV?

Response 9: A paragraph "Determination of HPV mediated Carcinoma" was added to the Material & Methods section.

Point 10:Page 3, lines 75-77: It is important to establish adequately the percentages of cell counting. Please review all the rates, due in the value “low” is not well understood what do the authors refer to with very few cells. ; Table 2. Is it necessary to include the term “general expression”? I suggest excluding it as possible.I suggest that less of 10% of the cell counting to be negative, and >10% could be considered positive. Table 2. Why do the authors include three values when they are only including positive and negative?

Response 10: Tumors with less than 10% stained cells were considered to be negative, >10% to be positive for a receptor, as you suggested. The term "General epxression" was excluded. All results and tables were adjusted according to the new threshold.

Point 11: Why do the authors define the term R(C)T in this section? What did they explain with R(C)T is radiotherapy+ radiotherapy and chemotherapy or separated?

Response 11: We added a definition of the term to our text. R(C)T is defined as Radio- or Chemotherapy without surgery. Page 7 line 197. Please specify the meaning of the term “alone”. Are the authors referring to no surgery treatments?

Point 12: Line 287. Why do the authors make this observation if they did not evaluate cancer stem cells?

Response 12: The term cancer stem cells was excluded and the discussion was completely re-written. We pointed out why TrkC was our main focus. 

Round 2

Reviewer 1 Report

The additional comments are provided in the response letter usingpurple color. 

Author Response

We added the background information to the introduction, as you requested. There are table legends associated with Table 1 and 2. Also, statistical analysis including p-values were added to Table 2 and Figure 2B. For Figure 2B, we provided a darker grey instead of black lines with different shapes because we think it is easier to distinguish. An additional interpretation of the difference between CD271 and Trk receptors in the molecular mechanisms was added to the Discussion. 

Reviewer 2 Report

Dear authors: Thank you for responding me all my doubts and answer my questions. I consider that manuscript improve and it is understandable. Thus I recommend the manuscript for publication. 

Author Response

Thank you for reviewing our manuscript. The introduction and discussion have been supplemented with some background information based on the suggestions of another reviewer.